# Survival of sentinel node biopsy versus observation in intermediate-thickness melanoma: A Dutch population-based study

R. M. H. Roumen[1,2]*, M. S. Schuurman[3], M. J. Aarts[3], A. J. G. Maaskant-Braat[1], G. Vreugdenhil[4], W. J. Louwman[3]

1 Department of Surgery, Máxima Medical Center, Eindhoven/Veldhoven, The Netherlands, 2 GROW–School for Oncology and Developmental Biology, Maastricht University Medical Centre, Maastricht, The Netherlands, 3 Netherlands Comprehensive Cancer Organization, IKNL, Utrecht, The Netherlands, 4 Department of Medical Oncology, Máxima Medical Center, Eindhoven/Veldhoven, The Netherlands

* r.roumen@mmc.nl

## Abstract

### Background

The Multicenter Selective Lymphadenectomy Trial (MSLT-1) comparing survival after a sentinel lymph node biopsy (SLNB) versus nodal observation in melanoma patients did not show a significant benefit favoring SLNB. However, in subgroup analyses melanoma-specific survival among patients with nodal metastases seemed better.

### Aim

To evaluate the association of performing a SLNB with overall survival in intermediate thickness melanoma patients in a Dutch population-based daily clinical setting.

### Methods

Survival, excess mortality adjusted for age, gender, Breslow-thickness, ulceration, histological subtype, location, co-morbidity and socioeconomic status were calculated in a population of 1,989 patients diagnosed with malignant cutaneous melanoma (1.2–3.5 mm) on the trunk or limb between 2000–2016 in ten hospitals in the South East area, The Netherlands.

### Results

A SLNB was performed in 51% of the patients (n = 1008). Ten-year overall survival after SLNB was 75% (95%CI, 71%-78%) compared to 61% (95%CI 57%-64%) following observation. After adjustment for risk factors, a lower risk on death (HR = 0.80, 95%CI 0.66–0.96) was found after SLNB compared to observation only.

### Conclusions

SLNB in patients with intermediate-thickness melanoma on trunk or limb resulted in a 14% absolute and significant 10-year survival difference compared to those without SLNB.

**Data Availability Statement:** Data cannot be shared publicly because of very strict privacy regulations of the Netherlands Comprehensive

Cancer Organization. Data are available from the Netherlands Comprehensive Cancer Organization to researchers who meet the criteria for access to confidential data (contact email: gegevensaanvraag@iknl.nl).

**Funding:** The author(s) received no specific funding for this work.

**Competing interests:** The authors have declared that no competing interests exist.

## Introduction

The sentinel lymph node biopsy (SLNB) is unequivocally a significant and reliable prognostic marker for cutaneous melanoma, especially in intermediate-thickness melanoma patients [1,2]. The final analysis of the Multicenter Selective Lymphadenectomy Trial (MSLT-1) of sentinel lymph node biopsy (SLNB) versus nodal observation in melanoma patients concluded that a SLNB in patients with intermediate-thickness provides accurate and important staging information, enhances regional disease control, and, among patients with nodal metastases, seems to substantially improve melanoma-specific survival [3]. The trial conclusions were called practice changing and SLNB was considered standard of care and critical for identifying patients eligible for adjuvant therapy and treatment trials [4]. The primary endpoint of the MSLT-I trial, however, was the overall 10-year melanoma-specific survival, which appeared not significantly different between both groups, being 81.4% versus 78.3% [3]. Due to this non distinctive primary outcome the value of performing a SLNB in melanoma patients still remains a subject of great debate [2,3,5,6].

The discussion on this matter can be summarized as between two hypothesis camps: those who consider lymph nodes to be *incubators* for sequential "orderly progression" of the disease and those who consider this only as *marker or indicator* of the melanoma's metastatic potential [7,8].

Until 2012 the Dutch guideline for management of melanoma did not routinely advise SLNB in daily clinical practice for melanoma of any Breslow thickness, but considered it an optional staging procedure [9,10]. As a consequence, some surgeons performed this procedure after obtaining informed consent, while others performed wide local excision only and advised follow-up. In 2016 this guideline was renewed and since then SLNB was routinely advised to patients with cutaneous melanoma > 1mm [11]. This type of selection bias in the Netherlands before 2016, may have resulted in a dichotomy of treatment due to the surgeon's and/or patient's preference. As a consequence, outcomes may have been quite different depending on local hospital policies [12].

In the Netherlands from 2019 onwards a positive lymph node >1mm allows for adjuvant immune- or targeted therapy. However, during the period described in the current study there was much discussion on the usefulness of performing SLNB, besides it being a valuable staging tool. The variation that occurred from surgeons that believed in the SLNB procedure, and those that did not, allows for this type of study, which will also be useful in interpreting the survival benefits of adjuvant treatments.

The aim of the present study was to evaluate whether performing a SLNB or not was associated overall survival in patients who were treated for intermediate-thickness skin melanoma in a population-based setting in the South East area of the Netherlands.

## Materials and methods

### Data and patients

All patients diagnosed with an intermediate thickness melanoma (1.2 mm– 3.5 mm Breslow thickness) in one of the 10 hospitals of the South East region of the Netherlands Cancer Registry (NCR) between 2000 and 2016 (n = 1989) were selected for this study. The cut-off for intermediate thickness melanomas is in accordance with the cut-offs used in the MSLT-trials, which enables comparison with existing literature. The cancer registry in this area was founded in 1955 and currently covers a population of 2.4 million inhabitants, ten general hospitals and six pathology departments. This population-based database includes all newly diagnosed cancer patients in the South East part of the Netherlands, and is now embedded within the Netherlands Comprehensive Cancer Organization.

This study was approved by the privacy committee (Commissie van Toezicht) of the NCR and all data were fully anonymized before data-analyses. Registration is primarily based on notification by the automated pathology archive and hospital discharge notes.

Patient, tumor and treatment characteristics are retrieved from patient files by specially trained registration clerks. Data quality is high due to thorough training of the registration clerks and by a variety of computerized consistency checks. Completeness is previously found to exceed 95% [13]. Classification of tumor characteristics is recorded according to the TNM Classification of Malignant Tumors [14], and by International Classification of Diseases for Oncology (ICD-O-3) [15]. The Netherlands Cancer Registry is annually linked to the Municipal Personal Records database to retrieve information on vital status and date of death. The follow-up data were completed until February 1st 2021. Data on gender, age, year and hospital of diagnosis, co-morbidity (according to a slightly adapted version of Charlson's Co-morbidity Index) [16], histological subtype of primary melanoma, presence or absence of ulceration, location of the primary tumor, treatment (excision with or without SLNB), results of the SLNB (positive/negative), vital status and date of death were extracted from the NCR for analyses. Patients with clinically suspect (palpable) lymph nodes, those who's clinical nodal status was not explicitly stated as negative, those who underwent some form of lymph node biopsy or dissection without a previous SLNB, or stage IV disease at primary melanoma diagnosis were excluded. Patients were considered to have undergone SLNB only if this was part of their initial treatment (within 6–9 months from initial melanoma diagnosis). Socioeconomic status was available at a small area-based level for each postal code, based on individual fiscal data on the economic value of the home and household income [17]. The study population was divided into two groups based on intention-to-treat during the first 6 months after their melanoma diagnosis: patients undergoing wide local excision only (Observation group) or those who received a wide local excision combined with a SLNB (SLNB group). A decision to perform either operation was entirely based on the surgeon's preference, on the patient's preference, or on a combination thereof. Patients were analyzed according to their initial grouping, irrespective of possible lymph node dissection later during the course of their disease. In the Netherlands (plastic) surgeons are hesitant to perform a SLNB in the head and neck region because of the alleged possible harmful consequences of this technically challenging procedure in this anatomic region. Moreover, sensitivity of the procedure in this area has been doubted because of high false negative rates [18–20]. We therefore excluded this group of patients in the present study, as was also done in the DeCOG study [21]. Moreover, in the 2016 guideline performing a SLNB in the head and neck region was still considered optional and not routine practice [11].

## Statistical analysis

Distribution of patient and tumor characteristics were tested using chi-square tests. Survival time was defined as time from diagnosis to death, or February 1, 2021 for those patients who were still alive. Survival curves were constructed and the log-rank test was used to compare the survival rates of patients with and without a SLNB. Multivariable Cox regression models were performed to assess the impact of performing a SLNB on overall survival. We included variables in the model according to stepwise forward regression analyses, using the Akaike Information Criterion to choose the best fitted model. We used the stepwise options for entering variables to the model, and we chose the model with the best fit. The factors included in the multivariable analyses were age, gender, Breslow thickness, tumor ulceration, location of the primary tumor, co-morbidity (number of concomitant diseases), socioeconomic status and period of diagnosis.

Analyses were performed using Stata Statistical software (version 13.1) and SAS version 9.4 (SAS Institute Inc., SAS Campus Drive, Cary, NC, USA). A p-value less than 0.05 was considered statistically significant.

## Results

Between January 2000 and December 2016, a total of 1989 patients with intermediate thickness melanoma were diagnosed and treated by either wide local excision only (Observation group, n = 981) or by wide local excision followed by a SLNB procedure (SLNB group, n = 1008). All surgical procedures were performed in 10 different hospitals (A–J) in the South East part of Netherlands. Number of new onset melanoma patients per hospital of diagnosis ranged from 125 to 312 during this 17 year time period. SLNB rates increased over time in the last 10 years (Fig 1A) (average 51%, range 32%-69%). Fig 1B illustrates the percentage of SLNB procedures done over the entire period per hospital, indicating the large variation in surgical treatment in the region.

Baseline characteristics of these two groups are summarized in Table 1. Significant differences were observed with respect to age at diagnosis (more elderly in the Observation group, p<0.001), and histological subtype (p<0.001). Moreover, patients from the Observation group had more co-morbidity and were more often of lower socioeconomic status (p<0.001 and p = 0.001, respectively).

Sentinel lymph nodes were positive in 19% of patients (n = 194). Predictive for the chance of undergoing a SLNB were: age, location of melanoma, number of co-morbidities, socio-economic status, period of diagnosis and hospital. In contrast, gender, Breslow thickness, presence of ulceration and histological subtype were not predictive. An additional full lymph node dissection was performed in 125 patients (64%) of all 194 patients with a positive sentinel node.

### Survival

Overall survival rates were significantly higher in the patients undergoing SLNB compared to the Observation group as the 10-year survival rates were 75% (95%CI: 71%-78%) and 61% (95%CI: 57%-64%), respectively (p<0.001) (Fig 2). Median follow up was 6.7 years (range: 0.1–20.0). After adjustment for age, gender, socioeconomic status, Breslow thickness, presence of ulceration, type and location of melanoma and co-morbidity in multivariable analysis, SLNB patients had a lower overall mortality risk as compared to those of the Observation group only, which was statistically significant (HR 0.80, 95% CI 0.66–0.96) (Table 2). No association with early or late period on the outcome of overall survival was observed and no differences in survival rates related to localisation were found when patients with melanoma of the trunk or limb were analysed separately.

### Post hoc external validation

After finding the results of these detailed analyses of the 10 hospitals of the Eindhoven region of the Netherlands Cancer Registry (NCR), we validated this within the nationwide Netherlands Cancer Registry, containing data on SLNB among melanoma patients of all Dutch hospitals since 2010. These data were less detailed (e.g. information on co-morbidity was unavailable at a nation-wide level) and had shorter follow up (median 4.6 years). Using the same in- and exclusion criteria, 8,274 patients were analyzed, those patients had similar age and gender distribution compared to our study population. Within this larger dataset we found that 5-year survival of patients with SLNB was 86% compared to those in the Observation group, being 72%, (p < 0.001).

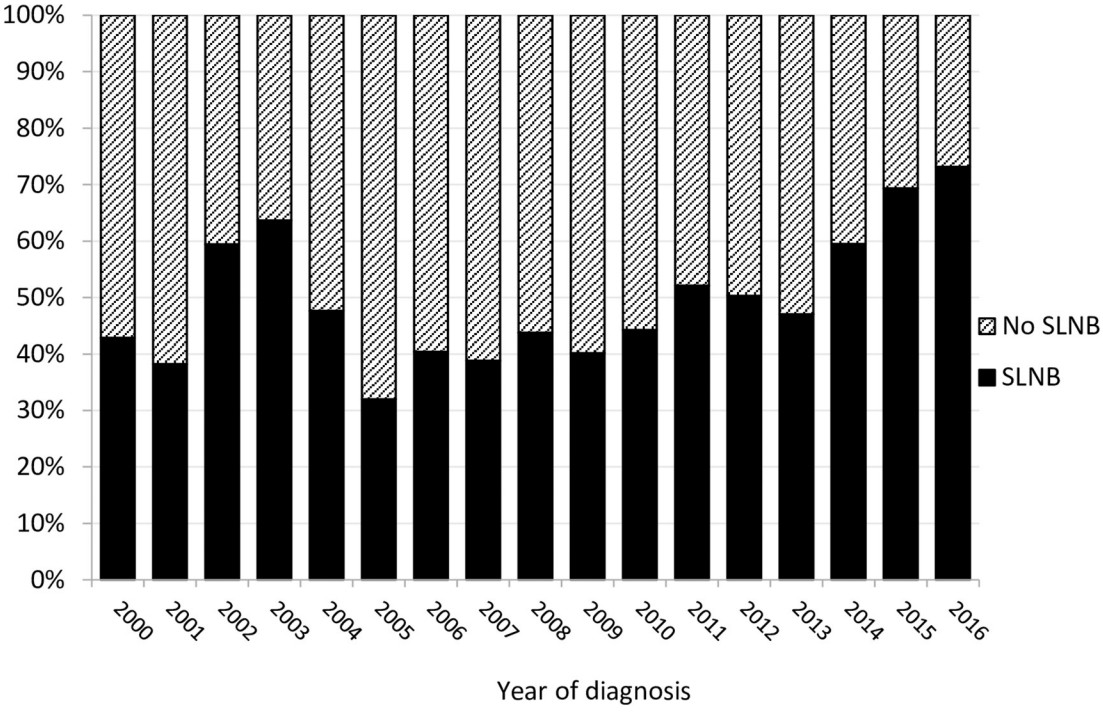

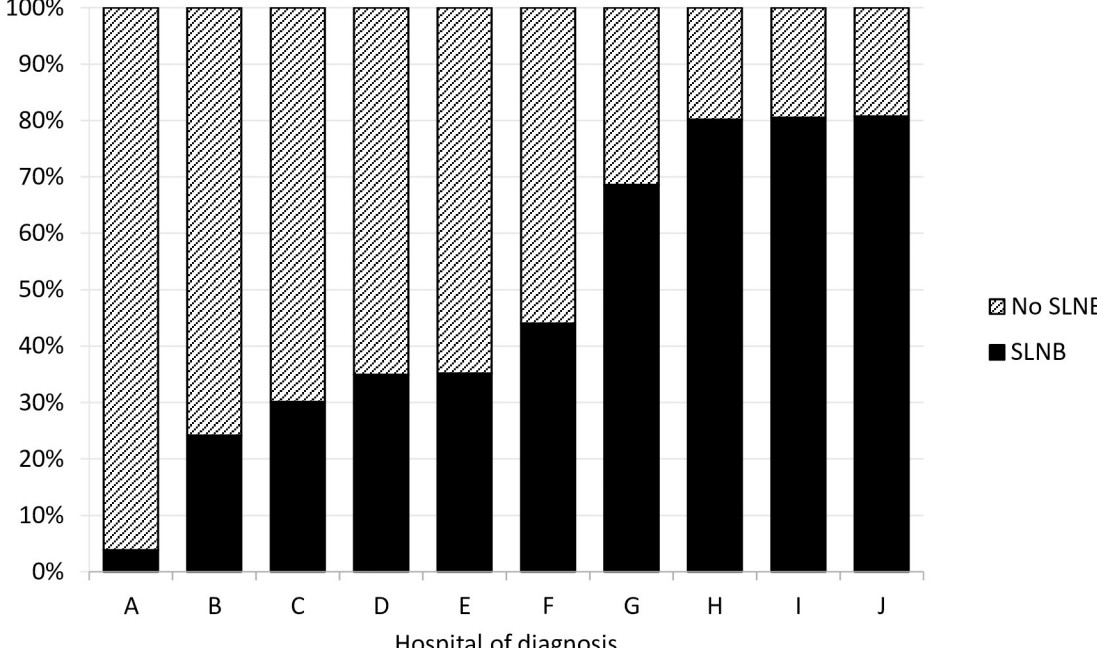

**Fig 1. Percentage of patients with intermediate-thickness (1.2–3.5 mm) melanoma on trunk or limb in Southern Netherlands who underwent SLNB (Sentinel Lymph Node Biopsy).** 1A by year of diagnosis 1B by hospital of diagnosis.

## Discussion

This population-based study shows that patients with intermediate-thickness melanoma treated by wide local excision ànd SLNB had a higher overall 10-year survival than patients without such a SLNB procedure and observation only. There was a 14% absolute survival

**Table 1. Characteristics of patients with intermediate thickness cutaneous melanoma (1.2–3.5 mm) on trunk or limb (n = 1989) diagnosed 2000–2016 in the Southern Netherlands.**

| | Observation | | SLNB[1] | | p-value |
|---|---|---|---|---|---|
| | N | % | N | % | |
| *Number of patients* | 981 | 49% | 1008 | 51% | |
| *Age at diagnosis* | | | | | |
| Median (Q1-Q3) | 61 | (49–72) | 54 | (44–66) | |
| <40 years | 109 | 11% | 172 | 17% | <0.0001 |
| 40–49 years | 134 | 14% | 210 | 21% | |
| 50–59 years | 221 | 23% | 233 | 23% | |
| 60–69 year | 220 | 22% | 212 | 21% | |
| ≥70 years | 297 | 30% | 181 | 18% | |
| *Gender* | | | | | 0.85 |
| Men | 479 | 49% | 488 | 48% | |
| Women | 502 | 51% | 520 | 52% | |
| *Breslow thickness* | | | | | |
| Median (Q1-Q3) | 1.76 | (1.4–2.3) | 1.88 | (1.4–2.5) | |
| 1.20–1.7 mm | 482 | 49% | 448 | 44% | 0.13 |
| 1.71–2.3 mm | 256 | 26% | 273 | 27% | |
| 2.31–2.9 mm | 139 | 14% | 173 | 17% | |
| 2.91–3.5 mm | 104 | 11% | 114 | 11% | |
| *Tumor ulceration[1]* | | | | | 0.55 |
| No | 676 | 69% | 705 | 70% | |
| Yes | 196 | 20% | 206 | 20% | |
| Unknown | 109 | 11% | 97 | 10% | |
| *Location of melanoma* | | | | | 0.62 |
| Arm | 238 | 24% | 229 | 23% | |
| Leg | 311 | 32% | 337 | 33% | |
| Trunk | 432 | 44% | 442 | 44% | |
| *Histological subtype* | | | | | <0.0001 |
| Superficial spreading melanoma | 522 | 53% | 633 | 63% | |
| Nodular melanoma | 172 | 18% | 189 | 19% | |
| Other/unknown/unspecified | 287 | 29% | 186 | 18% | |
| *Number of comorbidities* | | | | | <0.0001 |
| None | 520 | 53% | 573 | 57% | |
| 1 | 188 | 19% | 200 | 20% | |
| 2 | 193 | 20% | 126 | 13% | |
| Unknown | 80 | 8% | 109 | 11% | |
| *Socioeconomic status* | | | | | 0.01 |
| High | 357 | 36% | 391 | 39% | |
| Intermediate | 361 | 37% | 389 | 39% | |
| Low | 180 | 18% | 149 | 15% | |
| Institute | 31 | 3% | 14 | 1% | |
| Unknown | 52 | 5% | 65 | 6% | |
| *Period of diagnosis* | | | | | <0.0001 |
| 2000–2004 | 183 | 19% | 186 | 18% | |
| 2005–2007 | 209 | 21% | 124 | 12% | |
| 2008–2010 | 194 | 20% | 145 | 14% | |
| 2011–2013 | 241 | 25% | 240 | 24% | |

*(Continued)*

**Table 1.** (Continued)

| | Observation | | SLNB[1] | | p-value |
|---|---|---|---|---|---|
| | N | % | N | % | |
| 2014–2016 | 154 | 16% | 313 | 31% | |

SLNB = Sentinel Lymph Node Biopsy.

[1]information on tumor ulceration available from 2003 onwards.

difference, with curves still diverging 10 years after diagnosis. After adjustment for well-known melanoma related factors such as age, gender, Breslow-thickness, ulceration, histological subtype and location, co-morbidity and socioeconomic status, the risk of death after SLNB was significantly reduced (HR 0.80, 95% CI 0.66–0.96).

As far as we know, this is the third report showing a substantial survival benefit of the SLNB procedure per se in a large population-based daily clinical setting. Based on USA-SEER data of over fifty thousand patients, Chen et al [22] showed that patients who had undergone a SLNB had significantly longer 5-year overall survival rates compared to those who had not undergone this procedure (84.3% vs 70.1%, p = 0.000) [22]. These survival data are remarkably comparable with the present findings and external validation in the Dutch nationwide data base. However, Chen's study did not investigate whether SLNB was an independent factor. Murhta et al [23] also used SEER data from 2010–2012 and analyzed 13,703 cases, divided into thin, intermediate-thickness and thick, with only 16 months follow up. In this large cohort they showed after correcting for various factors, including socio-economic background, by multivariable analysis, that particularly in intermediate thickness and thick melanoma patients, SLNB was significantly associated with improved OS and MSS, with hazard ratio's of 0.46 and 0.608, respectively [23]. Unfortunately, they could not correct for comorbidity data, since

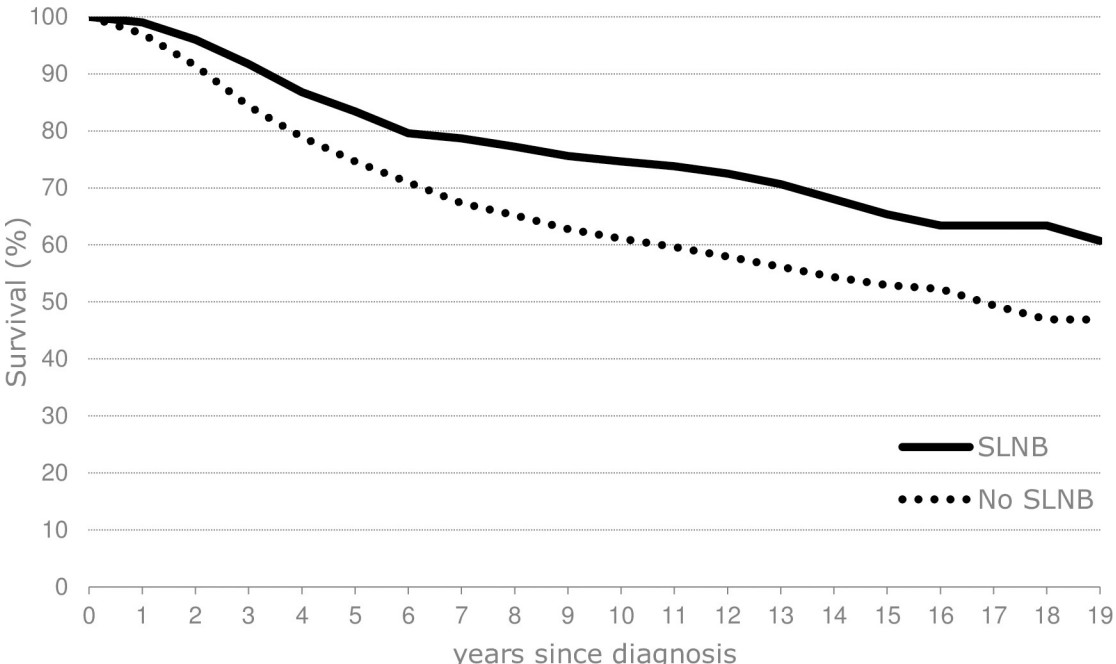

**Fig 2. Survival of patients with intermediate-thickness (1.2–3.5 mm) melanoma on trunk or limb in Southern Netherlands diagnosed 2000–2016 followed up until Feb 2021 by SLNB (Sentinel Lymph Node Biopsy).**

**Table 2. Hazard Ratios (HR) of patients with intermediate-thickness (1.2–3.5 mm) melanoma diagnosed 2010–2016, followed-up until February 2021.**

| | HR[1] | 95% CI |
|---|---|---|
| **SLNB[2]** | | |
| No | 1.00 | |
| Yes | **0.82** | **(0.69–0.96)** |
| **Age at diagnosis** | | |
| Continuous | **1.04** | **(1.04–1.05)** |
| **Gender** | | |
| Men | 1.00 | |
| Women | **0.79** | **(0.66–0.94)** |
| **Breslow thickness** | | |
| Continuous | **1.52** | **(1.34–1.71)** |
| **Tumor ulceration[3]** | | |
| No | 1.00 | |
| Yes | **1.31** | **(1.08–1.60)** |
| Unknown | 1.22 | (0.90–1.66) |
| **Location of melanoma** | | |
| Arm | 1.00 | |
| Leg | 0.84 | (0.67–1.05) |
| Trunk | 1.13 | (0.92–1.39) |
| **Number of comorbidities** | | |
| None | 1.00 | |
| 1 | 1.08 | (0.87–1.35) |
| 2 | **1.50** | **(1.20–1.88)** |
| Unknown | 1.20 | (0.71–1.47) |
| **Socioeconomic status** | | |
| High | 1.00 | |
| Intermediate | **1.27** | **(1.05–1.53)** |
| Low | **1.40** | **(1.12–1.76)** |
| Institute | **2.77** | **(1.86–4.11)** |
| Unknown | 0.87 | (0.55–1.37) |
| **Period of diagnosis** | | |
| 2000–2004 | **1.77** | **(1.26–2.51)** |
| 2005–2007 | **1.68** | **(1.23–2.29)** |
| 2008–2010 | **1.46** | **(1.06–2.00)** |
| 2011–2013 | **1.47** | **(1.10–1.98)** |
| 2014–2016 | 1.00 | |

[1] HR = Hazard Ratio, each variable adjusted for all other variables.

[2] SLNB = Sentinel Lymph Node Biopsy.

[3] information on tumor ulceration available from 2003 onwards.

these are lacking in SEER. Another large population-based paper from the Swedish Melanoma Register (also over 50,000 patients) unfortunately failed to mention the effect of performing the SLNB or not as separate factor for mortality [24]. Five more reports comparing SLNB versus observation were found and are summarized in Table 3 [18,25–28]. The conclusions are inconsistent, but various differences in favor of SLNB were observed after univariable analyses. However, these differences disappeared after multivariable analyses or were seen only in

**Table 3. Studies on melanoma comparing outcome of SLNB and excision versus Obs (observation) after excision only.**

| Author (year) | Melanoma thickness | Number of patients SLNB vs Obs | Type of analysis | Survival* OS/MSS | Adjustment for comorbidity | Adjustment for socioeconomic status |
|---|---|---|---|---|---|---|
| Möhrle [24] (2004) | 0.1–14 mm (50% <1 mm) | 271 vs 2617 | Historical control Multivariable | No OS difference SLNB vs Obs | No | No |
| Satzger [25] (2011) | Stage I/II ≥1 mm | 296 vs 377 | Historical control Univariable | 5y MSS better 84.8% vs 80.3% not tested multivariable | No | No |
| Van der Ploeg [26] (2014) | ≥1 mm | 2,909 vs 2,931 | Retrospective Univariable | 5y MSS 1–4 mm better 86.8% vs 85.3% not tested multivariable | No | No |
| Sabell [18] (2015) | ≥1 mm (only ≥75 years) | 340 vs 213 | Retrospective Univariable | OS better SLNB not tested multivariable | Yes | No |
| Chen [22] (2016) | ≥1 mm | 28,443 vs 18,908 | SEER Univariable | 5y OS better 84.3% vs 70.1% | No | No |
| Kim [28] 2016 | ≥0.75 mm (pediatric < 20 years) | 261 vs 49 | SEER Propensity matched | No MMS difference SLNB vs Obs at 84 mth | No | No |
| Murhta [23] 2018 | ≥0.75 mm Thin, intermediate and thick | 8,205 vs 5,498 | SEER Multivariable | SLNB significantly better OS HR^ 0.46 (CI: 0.384–0.551) and MSS HR 0.608 (CI: 0.420–0.881) | No | Yes |
| Present study: Roumen (2021) | 1.2–3.5 mm | 1,008 vs 981 | NCR# Univariable & Multivariable | 10y OS better 75% vs 61% also significant after adjustment | Yes | Yes |

*Survival: OS/MSS = Overall Survival/Melanoma Specific Survival.

#NCR = Netherlands Cancer Registry.

^HR = Hazard Rate.

specific subgroups. In none of these papers corrections were made for comorbidity or socio-economic class.

The present study is quite unique as it reports on a large series of intermediate-thickness melanoma patients representing daily clinical practice in the Netherlands until the end of 2016. Such an approach was possible as the former Dutch guideline until 2012, in contrast to the American, considered a SLNB procedure optional as was also long time proposed in the UK [10,29,30]. Nevertheless, also in the USA there is still large variation in performing a SLNB (and additional treatment) per county/state and even within one institute [18,31,32]. Such variations are seen in many countries all over the world and likely depend on age, sex, race, socio-economic status, region, surgeon's preference and even type of health care insurance [33–35]. In the Netherlands, this variation was also present, although access to healthcare was equal for each patient [12,36].

Several factors, such as age and co-morbidity, were found to play a role in the treatment selection process. These factors should be kept in mind when interpreting results of other studies that did not adjust for these items. In the present study we observed a considerable variation (4% to 81%) in SLNB hospital rates (Fig 1A). Although incidences of intermediate-thickness melanoma over the entire period increased, percentages of SLNB hovered around a mean of 50% (ranging from 32–81%), which is in accordance to other Dutch reports [12].

How do the present data relate to the landmark MSLT-I trial? In that trial, 1270 patients were randomized to SLNB (n = 770) or observation (non-biopsy) group (n = 500) [3]. The 10-years melanoma-specific survival was approximately 3% better in the biopsy group (81.4% +/- 1.5% versus 78.3% +/- 2.0%; p = 0.18), but possibly not statistically significant because the

sample size was too small. This difference was mainly caused by the dilutional effect of a supreme survival of approximately 20% of the population who had undergone immediate versus delayed lymphadenectomy. This survival benefit was still present when patients with false negative sentinel nodes were included in the analyses.

Although we analyzed overall survival and not melanoma specific survival, the 14% difference between the present groups is much larger than the 3% difference in MSLT-I. Several explanations for this latter difference may be suggested. First, MSLT-1 patients were on average younger compared to our Observation group, (52 vs 64 yr). Whether patients of the MSLT-I study were more vital (had less co-morbidity) or were of higher socioeconomic class, both factors associated with improved survival, is unknown [37,38].

Another possibility is that the daily clinical practice selection process identifying candidates unsuitable for a SLNB (observation only) has a far more negative impact on survival, compared to mortality in randomly trial patients. For instance, the discovery of regional nodal recurrence is possibly more delayed in daily clinical practice compared to a trial setting with strict follow up criteria. This bias can result in a larger lymph node tumor burden, which is known to have a negative impact on survival [39–41]. Moreover, patients within the present daily clinical setting who had a SLNB and positive nodes may have had possibly better access to alternative melanoma treatment strategies including adjuvant trials, or even better access to health care in general. In general, randomized controlled (cancer) trials participants may demonstrate improved survival compared to non-trial participants with a similar disease stage, a phenomenon known as the Hawthorne effect [42,43]. In conclusion, the routine selection process as commonly performed in a daily clinical setting may result in a negative impact on survival in patients not undergoing a SLNB procedure.

In 2016 and 2017 two RCT's (MSLT-II and DeCOG-SLT) on the role of completion lymph node dissection (LND) for SLN positive patients were published [21,44]. In both studies no difference in overall or melanoma specific survival could be demonstrated between patients who underwent completion LND versus those who had observation only. In both studies the majority of patients in both arms had only one positive SLN and low volume disease in these SLN (< 1mm). The rate of positive non-SLNs in the completion LND group was also low. This latter finding is a very strong predictor of distant recurrence and melanoma specific survival [45,46]. We speculate on the meaning of the combination of these trials with MSLT-I data and the present findings. In patients with low volume metastatic disease in SLNs without positive non-SLNs in the regional basin, removal of this tumor burden by SLNB can be therapeutic, supporting the incubator hypothesis. When tumor load in the regional lymphatic basin increases, the beneficial value of performing a therapeutic SLNB decreases because metastatic and distant disease has already developed, supporting the marker or indicator hypothesis. In this way both theories are complementary and not mutually exclusive.

The current study is probably limited by the presence of other unknown confounding factors, as is always the case in retrospective studies. However, by including age, co-morbidity and socioeconomic status we could adjust for a few very important factors. Next, detailed clinical follow up data after the diagnosis of positive nodal or metastatic disease were also not available nor evaluated. On purpose, we left out detailed information on the consequences of a positive or negative SLN, as this was not deemed relevant for our primary outcome question. There are enough data on this issue in melanoma literature. In the present series, until 2016, 64% of SLNB positive patients underwent an immediate completion lymph node dissection. We now know that such an adjuvant surgical procedure does not result in a significant survival benefit as clearly shown in the two above referred RCT's on this issue [21,44]. A strong point of the present analysis is real life daily clinical practice population-based data with a random selection of the investigated parameter (SLNB or not).

## Conclusion

In the past, in a Dutch population-based setting after multivariable correction for confounders, performing a SLNB was associated with a significant overall survival benefit in intermediate-thickness limb and trunk melanoma patients compared to observation after wide local excision alone. At present, performing a SLNB is mostly advised because of the impact on further adjuvant systemic (immune)therapy or trial participation. Moreover, the present data indicate that performing a SLNB in these patients has also some therapeutic effect, supporting in some way the incubator hypothesis [7].

## Author Contributions

**Conceptualization:** R. M. H. Roumen, A. J. G. Maaskant-Braat, G. Vreugdenhil, W. J. Louwman.

**Data curation:** R. M. H. Roumen, M. S. Schuurman, M. J. Aarts.

**Formal analysis:** R. M. H. Roumen, W. J. Louwman.

**Methodology:** M. S. Schuurman, M. J. Aarts, W. J. Louwman.

**Supervision:** R. M. H. Roumen, W. J. Louwman.

**Validation:** R. M. H. Roumen, M. J. Aarts.

**Writing – original draft:** R. M. H. Roumen, G. Vreugdenhil, W. J. Louwman.

**Writing – review & editing:** R. M. H. Roumen, A. J. G. Maaskant-Braat, W. J. Louwman.

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
