## [Decision Letter · Decision Letter 0]

15 Oct 2020

PONE-D-20-28185

Survival of sentinel node biopsy versus observation in intermediate-thickness melanoma: a Dutch population-based study

PLOS ONE

Dear Dr. ROUMEN,

Thank you for submitting your manuscript to PLOS ONE. After careful consideration, we feel that it has merit but does not fully meet PLOS ONE’s publication criteria as it currently stands. Therefore, we invite you to submit a revised version of the manuscript that addresses point by point the concerns raised during the review process.

We look forward to receiving your revised manuscript.

Kind regards,

Anna Sapino

Academic Editor

PLOS ONE

Journal Requirements:

2. Please note that PLOS does not permit references to “data not shown.” Authors should provide the relevant data within the manuscript, the Supporting Information files, or in a public repository. If the data are not a core part of the research study being presented, we ask that authors remove any references to these data.

3. In your ethics statement in the manuscript and in the online submission form, please provide additional information about the patient records used in your retrospective study. Specifically, please ensure that you have discussed whether all data were fully anonymized before you accessed them.

5. Please include your tables as part of your main manuscript and remove the individual files. Please note that supplementary tables should be uploaded as separate "supporting information" files.

Reviewers' comments:

Reviewer's Responses to Questions

**Comments to the Author**

1. Is the manuscript technically sound, and do the data support the conclusions?

Reviewer #1: Yes

Reviewer #2: Partly

2. Has the statistical analysis been performed appropriately and rigorously? 

Reviewer #1: No

Reviewer #2: Yes

3. Have the authors made all data underlying the findings in their manuscript fully available?

Reviewer #1: Yes

Reviewer #2: Yes

4. Is the manuscript presented in an intelligible fashion and written in standard English?

Reviewer #1: Yes

Reviewer #2: Yes

5. Review Comments to the Author

Reviewer #1: I have reviewed with pleasure the paper by Roumen and coauthors. This large study looks at the survival of melanoma patients stratifying them in the basis of SLN. Despite the topic is not new, the procedure is still discussed and new literature data are always welcome. The sample is quite large and the paper is well written.

I suggest in order to improve the strength of the message few questions:

The definition of intermediate melanoma 1-3.5 mm, despite being used in MSLT-I is quite confusing, in fact it does not include the 3.5-4 mm melanomas, that are still considered intermediates, according to the AJCC. Please discuss the reason of this decision.

Statistic: I would suggest to perform an AKAIKE test to value is the model is correct

How the variables selection have been performed before the inclusion in the final model?

Have the authors considered the different therapeutical approach when including the variables in the final model?

Reviewer #2: The authors report a retrospective study of melanoma patients treated by SLNB or nodal observation in the period 2010-2106 with the objective to define whether SLNB could have an impact on overall survival. The adoption of SLNB for the treatment of melanoma is widely recognized in guidelines from all over the world, also considering that new adjuvant treatments can be performed only in the presence of SLNB positivity. The retrospective design of the study and the fact that the two patient groups are selected only on the basis of surgeon or patient decision to perform or not SLNB largely reduces the statistical significance of the results. Overall, the significance and clinical impact of these data in the daily practice is low as the discussion now is not whether to perform or not SLNB but rather whether to perform or not completion dissection.

Synopsis: in the daily clinical practice… the finding of a significant 10-year survival is not related to the daily clinical practice, rather influences daily clinical practice supporting the adoption of SLNB

Introduction, line 82: Due to this non distinctive primary outcome the value of

83 performing a SLNB in melanoma patients still remains a subject of great debate. Even considering the lack of 10-year survival differences between SLNB and nodal observation, SLNB biopsy represents standard of care in melanoma patients , thus its values can not be considered subject of great debate, also based on the possibility according to new data, to perform adjuvant treatments only in the presence of SLNB positive results. The authors should comment on it.

Line 84: this point is crucial but this distinction is relevant when considering completion dissection vs observation in SLNB positive patients rather than SLNB biopsy versus nodal observation in primary melanoma patients. The authors should comment on this issue.

Materials and Methods: line 105: 1.2 mm – 3.5 Breslow thickness : even if this Breslow interval was that chosen in the MSLT trial, nevertheless as it does not have a specific stage definition in TNMB classification , its relevance in the daily practice is questionable considering the time lapsed from the report of MSLT 1 results.

Line 122: The follow-up data were completed until January 1st 2018. Follow-up should be updated to 2020.

Line 137: The study population was divided into two groups: patients undergoing wide local excision only (Observation group) or those who received a wide local excision combined with a SLNB (SLNB group). This is a crucial point for the study as this statement introduces a wide range of biases and heterogeneity in the patient characteristics between these two groups that significantly reduces the statistical significance of the results. The authors should carefully discuss this issue and statistical adequate analyses to overcome this bias should be considered. Moreover, as the authors state that since 2016 SLNB biopsy even in Netherlands has become standard o f care, the relevance of these results is potentially clinically limited.

RESULTS line 175: patients in the observation group were elderly and with more comorbidities , thus highlighting the differences between the two groups .

Line 185: the percentage of patients undergoing full lymph node dissection after a positive SLNB was apparently lower than expected (64%) , the authors should comment on this issue.

Line 188 Survival , the authors considered overall survival which could be a controversial finding considering that the groip of patients undergoing only nodal observation is characterized by an elderly age. Could the authors provide data on disease-specific survival ?

Line 202: post hoc external validation: the control group of the Netherlands Cancer Registry should be described in terms of clinico-pathologic characteristics and compared to identify potential differences with respect to that of the Eindhoven region. Considering overall survival rates, the survival of SLNB patients in the Eindhoven group (75%) was similar to that of the observation group in the Netherlands Cancer Registry (72%) thus implying that the impact of SLNB procedure is definitely debatable based on these data.

It should be considered the survival of SLNB positive patients and compared with that of patients diagnosed with nodal metastases after observation.

6. PLOS authors have the option to publish the peer review history of their article (what does this mean?). If published, this will include your full peer review and any attached files.

Reviewer #1: No

Reviewer #2: No

---

## [Author Response · Author response to Decision Letter 0]

5 May 2021

see attached response to reviewers

---

## [Editor Report · Decision Letter 1]

10 May 2021

Survival of sentinel node biopsy versus observation in intermediate-thickness melanoma: a Dutch population-based study

PONE-D-20-28185R1

Dear Dr. Roumen,

We’re pleased to inform you that your manuscript has been judged scientifically suitable for publication and will be formally accepted for publication once it meets all outstanding technical requirements.

Kind regards,

Anna Sapino

Academic Editor

PLOS ONE
---

## [Editor Report · Acceptance letter]

17 May 2021

PONE-D-20-28185R1 

Survival of sentinel node biopsy versus observation in intermediate-thickness melanoma: a Dutch population-based study 

Dear Dr. Roumen:

I'm pleased to inform you that your manuscript has been deemed suitable for publication in PLOS ONE. Congratulations! Your manuscript is now with our production department. 

Kind regards, 

on behalf of

Dr. Anna Sapino 

Academic Editor

PLOS ONE